# Sustainable Development and Customer Satisfaction and Loyalty in North Cyprus: The Mediating Effect of Customer Identification

**Mustafa Ozkan [1], Kemal Cek [2,\*] and Serife Z. Eyupoglu [1]**

[1] Department of Business Administration, Near East University, Nicosia 99010, Cyprus; mustiozk@hotmail.com (M.O.); serife.eyupoglu@neu.edu.tr (S.Z.E.)

[2] Department of Accounting and Finance, Cyprus International University, Mersin 10, Nicosia 99258, Turkey

[\*] Correspondence: kcek@ciu.edu.tr

**Abstract:** This study analyzes the influence of corporate social responsibility (CSR) from a multi-dimensional sustainable development approach on customer loyalty and satisfaction through the mediation of customer identification. CSR is a key concept for banks to attract, retain, satisfy and increase the loyalty of customers. Structural equation modeling was conducted to test the proposed relationship. The questionnaire was filled out by 389 banking sector customers. The findings suggested that customer identification mediates the relationship between the economic development dimension of CSR and customer satisfaction and loyalty. However, customer identification did not show a mediating effect between social equity and environmental protection dimension of CSR and customer loyalty and satisfaction. The findings are expected to provide insights into the importance of CSR for the banking industry in order to enhance favorable customer attitudes.

**Keywords:** corporate social responsibility; customer loyalty; customer satisfaction; customer identification; banking industry

## 1. Introduction

In the changing and challenging world of business, which has become increasingly competitive over the past years, the price and quality of the products alone may not be sufficient to attain a competitive advantage over your competitors. The financial crisis has caused a loss of trust of customers in the banking sector and harmed the images of banks in their eyes [1]. The United Nations has placed emphasis on global environmental and sustainability concerns and adopted 17 sustainable development goals. This has increased the importance of aspects that can create a way for companies to be in a better competitive position. Considering the demands of stakeholders is one major way of achieving an advantage. Stakeholders of companies such as consumers, investors, governments, employees, and suppliers are now placing significant attention on the environmental and social aspects. Stakeholders exert pressure on the firms to engage in CSR activities [2]. In particular, consumers expect authentic and innovative experiences from any business they engage with [3]. Corporate social responsibility (CSR) has emerged as a concept that shows how companies respond to the demands of the stakeholders [4]. It is suggested that customers develop positive perceptions and behaviors towards socially responsible companies [5]. Previous studies have shown that CSR provides various benefits for companies in terms of profit, image, reputation, employee performance, customer satisfaction, and customer loyalty [5–7]. Therefore, firms should recognize and understand the needs of the customers in order to shape their behaviors. Furthermore, the banking industry signed an agreement called the Equator Principles, which proposes that signatory banks such as Bank of America, Barclays, HSBC, and ING support socially responsible development. In addition, CSR initiatives improve corporate reputation which favors consumer retention. Given that customers have no alternative that directly affects the company profit, they are the most limited asset for companies. In that sense, customer loyalty and satisfaction are crucial for

companies in achieving long term success. The global and financial crises make customer loyalty even more crucial [8]. While some studies have found a significant and positive relationship between CSR and loyalty [9], some studies have shown insignificant or even negative relationships [10]. Furthermore, customer identification acts as a driver between CSR and customer behavior [5,11]. This study proposed that CSR influences customer satisfaction and loyalty through the mediating effect of customer identification [7,12,13].

The current study adds to the literature in certain ways. Firstly, CSR awareness is growing, and customers are valuing it in emerging nations [14]. However, the majority of the research has focused on developed countries, and only a limited number of studies have been conducted in emerging nations [12,15–17]. This is important as the CSR activities in developing countries show contrasts with the activities in developed countries [18]. This study contributes to the literature by investigating the context of Northern Cyprus, which is an emerging country where the current study was carried out. Secondly, there is limited research on the CSR and customer-specific outcomes that focus on the banking industry, which is the industry with the highest contribution to social responsibility initiatives. It is suggested that the key feature of CSR within the banking industry includes financial complexity, risk management, business ethics, and consumer rights [19]. Banks seek out new marketing methods to increase service performance and attain consumers' favorable attitudes [20]. Banks have a dynamic role in generating economic and societal growth and development of countries [21]. The financial system of North Cyprus is based on banks as the primary institutions for financial activities, and the banking industry is considered to be one of the most critical industries that contribute to the overall development [22]. Lastly, sustainable development is crucial for emerging nations. It is suggested that sustainable development includes economic, environmental, and social dimensions [17]. It has been criticized that previous research used a one-dimensional approach to CSR when analyzing its effect on customer-related outcomes [23]. Thus, this study contributes to the literature by considering a multidimensional CSR conceptualization and using a multidimensional scale that is based on the sustainable development dimensions [24–26]. This study aims to evaluate the influence of sustainable development dimensions of CSR (economic, social, and environmental) on customer loyalty and satisfaction mediated by the effect of customer identification.

## 2. Theoretical Framework and Hypotheses Development

### 2.1. Corporate Social Responsibility

According to Aguinis [4], CSR can be defined as the organizational initiatives that consider the needs of the stakeholders, which include the economic, social, and environmental aspects. It includes the practices that are not made compulsory by authorities and are not carried out in the interests of the organizations. Even though many definitions have been suggested by scholars, there is no universally approved single definition of CSR. However, in all explanations, it is seen as an attempt to give back to nature and the community. CSR acts as a bridge that connects companies and the stakeholders through socially responsible activities that contribute to the interests of all the stakeholders [18]. Conceptualization of CSR is difficult due to the changing notions of the economic, environmental, and social expectations and the discretionary expectations.

CSR acts as a link between the stakeholders and organizations, such as employees and consumers. Stakeholders develop both positive and negative perceptions about the companies, and the importance of CSR in this can be explained by Freeman's stakeholder theory [27]. This theory suggests that organizations should consider the needs of all of their stakeholders, including employees, customers, and the community [27]. Therefore, stakeholders of an organization can develop more positive perceptions of the organization, which will bring benefits [17]. Corporations can end up in an unfavorable competitive position if they fail to fulfill stakeholders' societal expectations. Thus, many companies prioritize CSR, which has a positive effect on customer loyalty and purchase [7,15,28]. The conceptualization of CSR has two main ideas within the literature [24]. The first school

of thought suggested that the responsibilities of a company are focused on maximizing profits to create value for shareholders. A recent study concluded that firms still have a shareholder-focused approach in their CSR disclosures [29]. In addition, it is suggested that CEOs and managers have the incentives to motivate them to proactively participate in CSR [30]. The second school of thought suggests that the responsibility of a company is to consider the needs of all the stakeholders (Jones, 1980). In that sense, the sustainable development perspective is largely neglected by the literature, which calls for overall economic wealth, societal equity, and protection of the environment [23,24]. Considering a sustainable development perspective, CSR is the implementation of the principles of sustainable development at the organizational level. Thus, organizations have a commitment to achieving long-term and sustainable economic, social and environmental performance [24,31]. This study considered CSR as a multidimensional construct with economic, social, and environmental dimensions [26]. Stakeholder expectations that the firm will be profitable in the long-term, achieve higher rates of employment, and become efficient and effective in their business operations are classified as the economic dimension [24]. The association between the firm and the social environment is called the social dimension, which includes the responsiveness of the firm to social issues [25]. Firms' environmentally friendly activities have been classified as the environmental dimension, which includes resource usage, pollution reduction, and promoting environmentally innovative products/services [25].

### 2.2. Corporate Social Responsibility and Customer Identification

CSR is a significant tool for companies to generate identification and belongingness amongst their stakeholders. It is claimed that it enhances consumer trust and satisfaction [7]. Customer identification is one of the key implications of developing positive customer perceptions related to companies' actions [32]. The effect of perceived CSR on customer behaviors and attitudes has been researched by a number of studies that indicate customers can develop identification with the company. Previous research has evaluated the influence of CSR on the identification of customers with the company and found a positive influence [8,12,13]. Once a client identifies with an organization, they tend to show favorable behaviors. In the banking industry, a positive relationship between customer identification and purchase intentions has also been found [33]. According to the social identity theory and self-categorization theory, CSR helps companies to reflect an image of the company which is positively appealing to stakeholders, including customers, which leads to a process of categorization [34]. In turn, customers perceive this image of the companies and develop an identification [35]. The banking industry plays a key role in addressing today's need for financing to meet people's growing demands. The number of bank branches is steadily expanding in this regard [28]. The efficiency and effectiveness of banks are critical for the banking sector and the whole economy. Managing the performance of banks includes considering their environment, which can help banks to achieve a competitive advantage in the industry. Perceived CSR help companies to develop positive customer perceptions which enhance customer identification [8,36]. Therefore, the following hypotheses are proposed:

**Hypothesis 1 (H1a):** *Social CSR has a positive influence on customer identification.*

**Hypothesis 1 (H1b):** *Environmental CSR has a positive influence on customer identification.*

**Hypothesis 1 (H1c):** *Economic CSR has a positive influence on customer identification.*

### 2.3. Mediating Effect of Identification between CSR and Loyalty and Satisfaction

In the context of banking customers, it is suggested that the perceived CSR can lead to customer identification through positive emotions, and satisfaction can be achieved, which can promote loyalty [8].

Al-Ghamdi and Badawi [15] conducted research on the Saudi banking industry and found a positive correlation between CSR, loyalty, and satisfaction. Customer satisfaction,

which can be defined as the fulfillment of expectations of consumers with a service or a product, is crucial for a long-term successful relationship with customers [37]. According to the previous literature, customers who are satisfied with their chosen brand are highly likely to re-use its products (or services) and eventually turn into loyal customers. Customer satisfaction alludes to the fulfillment of the client's assumption [38]. This is an insight a client has in the wake of utilizing a specific item or administration, which precursors may originate from one or the other feeling or cognizance [38]. Furthermore, customer dissatisfaction can cause negative consequences for a company, such as negative feedback, word-of-mouth, and changing behavior. It is stated that negatively perceived CSR can result in lower customer satisfaction [7]. Customers as cognitive beings develop perceptions about companies and result in customer satisfaction. Firms disclose information about social responsibility initiatives which reduces the information asymmetry and uncertainty. Perceptions of CSR can result in higher customer satisfaction [8]. In addition, higher levels of identification are found to be a significant contributor to loyalty and positive feedback, and customers are found less likely to be affected by negative information [35].

It is proposed that customers have increased intention to revisit or repurchase from the company which they had experienced a positive involvement with. The tendency to engage in repurchasing or revisiting can adhere to loyalty when it occurs. Customers are the most important and limited assets of companies, and their loyalty is crucial for a company's success. Customer loyalty is defined as a customer's intention to revisit or rebuy a product or service [37]. Customer attraction is also increased as CSR activities are positively perceived [39]. Extra-role behaviors might be observed when the customers develop company identification. Being loyal to the company and increased satisfaction with the products/services of that company are examples of these behaviors [32]. CSR is found to be a direct and indirect contributor to customer loyalty [40]. In a recent study, loyalty and word of mouth were found to be significant factors affecting the customers' shopping attitudes towards retailers in Spain [3]. In addition, it is suggested that CSR positively affects the loyalty and satisfaction of hypermarket customers in Spain [41]. CSR is found to be an aspect that contributes to the quality and image of the firms, which, in turn, contributes to customer satisfaction and loyalty [41]. Similarly, e-service quality is found to be a significant contributor to e-satisfaction, which significantly affects the e-loyalty of fashion customers in Spain [42].

Consumers are now more concerned with the ethical and socially responsible behavior of companies they are engaging with. Thus, customers tend to develop satisfaction and loyalty to these companies [43]. For instance, Abbas, Gao [44] found that CSR can promote favorable customer behaviors such as loyalty, word of mouth, and feedback through customer engagement. In addition, Raza, Rather [12] found that there is a mediating effect of CSR on customer loyalty in Pakistan. CSR helps to achieve customer satisfaction by enhancing brand image [16]. However, there is still a need for alternative links between CSR and loyalty. While in some sectors CSR has been found to be an important player in customer loyalty (such as banks), in some sectors, such as hospitality, the effect of CSR has been insignificant [45]. These findings provide support to the idea that there can be mediating factors affecting the relationship.

Customer identification helps customers to become attracted and appeal to a company [32]. This affects their loyalty and satisfaction with a company. The identification generates a commitment that is more stable and long-term; thus, customer loyalty is achieved [32]. Customers not only identify with the product or services, but they also identify with the company, which is a stronger identification [40]. In other words, CSR increases the company's strength of identity and thus, enhances satisfaction and loyalty. Therefore, it can be argued that investing and participating in CSR initiatives can enhance profitability and competitiveness. This enhances the relationship with the consumer and company identification. Therefore, it results in an increase in consumer satisfaction and loyalty to the company [38,46]. It is suggested that positive feedback and repurchase intentions of customers tend to enhance customer identification with the company. Thus,

enhanced customer identification establishes customer loyalty and satisfaction in the long term [13,36,47]. Therefore, the following hypotheses are proposed:

**Hypothesis 2 (H2a):** *Customer identification mediates the relationship between social CSR and satisfaction.*

**Hypothesis 2 (H2b):** *Customer identification mediates the relationship between environmental CSR and satisfaction.*

**Hypothesis 2 (H2c):** *Customer identification mediates the relationship between economic CSR and customer satisfaction.*

**Hypothesis 3 (H3a):** *Customer identification mediates the relationship between social CSR and loyalty.*

**Hypothesis 3 (H3b):** *Customer identification mediates the relationship between environmental CSR and loyalty.*

**Hypothesis 3 (H3c):** *Customer identification mediates the relationship between economic CSR and customer loyalty.*

In light of the recent literature, the following research model is presented in Figure 1.

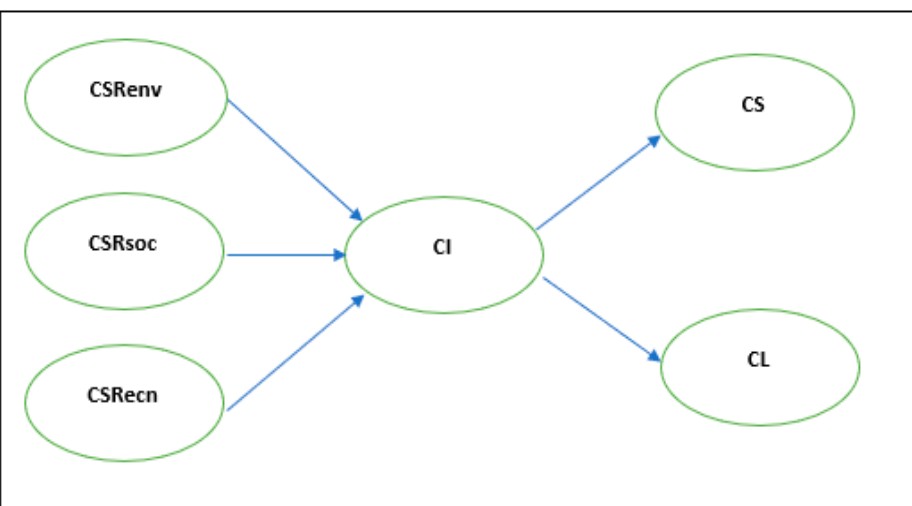

**Figure 1.** Research model. Notes: CSRsoc, social dimension of corporate social responsibility; CSRecn, economic dimension of corporate social responsibility; CSRenv, environmental dimension of corporate social responsibility; CS, customer satisfaction; CI, customer identification; CL, customer loyalty.

### 3. Methodology

In order to test the hypotheses, a quantitative method was used. Data have been collected from bank customers in North Cyprus as it suggested that banking industry is the main industry in which CSR needs to be investigated [48]. The population of the study in North Cyprus is approximately 286,257 in the last census. Thus, 384 respondents are satisfied to test the hypotheses in whole population at a 95% confidence interval level [49]. Initially, the questionnaire was translated into Turkish language by the authors and tested by a pilot study to test the validity and reliability of the scales. Due to the restrictions as a result of the COVID-19 pandemic, the survey was administered electronically between September and December 2020. Local banks have been selected in order to ensure that they adopt similar levels of CSR activities. The questionnaires have been delivered to customers of all 15 local banks operating in North Cyprus through online channels. The sample of the study consisted of randomly selected bank customers, and the questionnaires were sent to these customers as respondents. The banks were selected based on the criteria that they have at least one branch in each city of North Cyprus, and the respondents

were selected based on having a bank account at one of the local banks in North Cyprus. Accordingly, 389 valid questionnaires have been collected during this period which is a sufficient number of responses to achieve 95% confidence level as discussed above. A cover letter was included, which explains the aim, anonymity, and confidentiality of the questionnaire to decrease the social desirability bias. Non-response bias can cause biased estimates. Thus, to test the non-response bias, t-tests were conducted between early respondents and late respondents, which showed insignificant differences and bias, and hence is not an obstacle in this research [50].

Respondents identified whether they agree or disagree with a five-point Likert scale ranging from one (strongly disagree) to five (strongly agree). The questionnaire consisted of five sections, including demographics, CSR, customer identification, customer satisfaction, and customer loyalty. A 23-item CSR scale used in this study was developed by Alvarado-Herrera, Bigne [24]. This scale consists of three CSR dimensions: economic, environmental, and social. It supports the multidimensionality of the CSR construct. It supports the triple bottom line of sustainable development model [24]. An example item is "our bank is trying to sponsor cultural programmes". Customer satisfaction is measured by the satisfaction scale developed by Oliver [51], which includes items such as "I am satisfied with my decision to use this bank". Customer loyalty scale used in this study was developed by Zeithaml, Berry [52]. An example item is "I would do more business with this bank in the future". Customer identification used in this study is developed by Mael and Ashforth [53] scale developed to measure the customers' identification with the companies. An example item is "this organization successes are my successes".

## 4. Results

Initially, the demographic profiles of the respondents were analyzed. These included age, gender, education, and marital status. A total of 58% of the respondents were female, and 42% were male. A total of 25% of the respondents were aged between 18 and 25, 31% were aged between 25 and 40, 37% were aged between 40 and 65, and 7% were aged above 65. A total of 16% of the respondents were high school graduates. A total of 51% held an undergraduate degree. A total of 23% had a postgraduate degree, and 7.5% held a doctoral degree. A total of 50% of the respondents were married, 42% were not married, and 8% answered as other. Confirmatory factor analysis (CFA) was conducted using AMOS 21 software to test the factor loadings and construct validity. The factor loadings of the scale items are presented in Table 1 below. The CSR scale used in this study consisted of 23 items, of which 15 of these were significant and had loadings above the 0.5 threshold. The customer satisfaction scale consisted of four items, of which four of these have standard loadings above the 0.5 threshold. The customer loyalty scale consisted of five items, which were all found to be above the 0.5 threshold. The customer identification scale consisted of six items, two of which were at an acceptable level. The goodness of fit of the model was evaluated using: the comparative fit index (CFI), the goodness of fit index (GFI), the chi-square mean/degree of freedom (CMIN/df), the root means square error (RMSEA), and the standardized root mean square residual (SRMR). The threshold points for a good model fit should have CFI and TLI above 0.90, RMSEA below 0.05, and SRMR below 0.09 [54].

It is suggested that control variables that are not significant be excluded from the model, and hence this was the case in the structural model. The cut-off criteria for model fit indices were found to be acceptable for the model ($\chi^2$ (220) = 563.78, $p < 0.05$, comparative fit index (CFI) = 0.97, Tucker Lewis index (TLI) = 0.97, root mean square error of approximation (RMSEA) = 0.05, and standardized root mean square residual (SRMR) = 0.04). In addition, the discriminant validity was checked by considering the MSV and the squared correlation between the variables. Common method variance has been tested for the constructs. The single latent variable method was employed, and the results showed that common method variance is not an obstacle in this research.

**Table 1.** Scale items, standard loadings, standard error and significance.

|  |  | **Estimate** | **S.E** | *p* |
|---|---|---|---|---|
| CSR-Soc | CSR1 | 0.879 | 0.035 | *** |
|  | CSR2 | 0.890 | 0.037 | *** |
|  | CSR3 | 0.728 | 0.053 | *** |
|  | CSR4 | 0.850 | 0.055 | *** |
|  | CSR5 | 0.792 | 0.0.57 | *** |
|  | CSR6 | 0.800 | 0.058 | *** |
| CSR-Env | CSR7 | 0.883 | 0.056 | *** |
|  | CSR8 | 0.898 | 0.059 | *** |
|  | CSR9 | 0.894 | 0.056 | *** |
|  | CSR10 | 0.863 | 0.057 | *** |
|  | CSR11 | 0.792 | 0.058 | *** |
| CSR-Ecn | CSR12 | 0.713 | 0.056 | *** |
|  | CSR14 | 0.736 | 0.052 | *** |
|  | CSR15 | 0.838 | 0.053 | *** |
|  | CSR16 | 0.843 | 0.041 | *** |
| CI | CI1 | 0.884 | 0.050 | *** |
|  | CI2 | 0.786 | 0.055 | *** |
| CL | CL1 | 0.764 | 0.051 | *** |
|  | CL2 | 0.866 | 0.056 | *** |
|  | CL3 | 0.844 | 0.058 | *** |
|  | CL4 | 0.843 | 0.067 | *** |
|  | CL5 | 0.877 | 0.69 | *** |
| CS | CS1 | 0.917 | 0.035 | *** |
|  | CS2 | 0.898 | 0.036 | *** |
|  | CS3 | 0.834 | 0.037 | *** |
|  | CS4 | 0.851 | 0.045 | *** |

Notes: CSRsoc, social dimension of corporate social responsibility; CSRecn, economic dimension of corporate social responsibility; CSRenv, environmental dimension of corporate social responsibility; CS, customer satisfaction; CI, customer identification; CL, customer loyalty; S.E, standard error; *p*, significance; ***, significant.

Initially, the data analyses consisted of reliability and validity assessments. Cronbach's alpha coefficients were used to test the reliability of the scales. The coefficients ranged between 0.9 and 0.95, which is satisfactory. Furthermore, AVE was used to test for convergent validity and composite reliability to test for reliability; MSV is used for discriminant validity. According to Hair, Black [54], AVE should be higher than 0.50 in order to achieve convergent validity; CR should be higher than 0.70 in order to achieve reliability and to achieve discriminant validity, MSV should be lower than AVE. The results are shown in Table 2 below. The reliability and validity criteria were met, and the variables had significant correlations with each other.

**Table 2.** Correlation, reliability, and validity measures of the constructs.

|  | **CR** | **AVE** | **MSV** | **MaxR(H)** | **CSRsoc** | **CI** | **CSRecn** | **CSRenv** | **CS** | **CL** |
|---|---|---|---|---|---|---|---|---|---|---|
| **CSRSoc** | 0.917 | 0.689 | 0.638 | 0.927 | **0.830** |  |  |  |  |  |
| **CI** | 0.828 | 0.707 | 0.039 | 0.862 | 0.134 | **0.841** |  |  |  |  |
| **CSRecn** | 0.865 | 0.616 | 0.497 | 0.875 | 0.698 | 0.191 | **0.785** |  |  |  |
| **CSRenv** | 0.938 | 0.751 | 0.638 | 0.942 | 0.799 | 0.095 | 0.705 | **0.867** |  |  |
| **CS** | 0.944 | 0.809 | 0.154 | 0.945 | 0.392 | 0.165 | 0.369 | 0.280 | **0.899** |  |
| **CL** | 0.936 | 0.745 | 0.179 | 0.940 | 0.423 | 0.198 | 0.404 | 0.297 | 0.789 | 0.870 |

Notes: CSRsoc, social dimension of corporate social responsibility; CSRecn, economic dimension of corporate social responsibility; CSRenv, environmental dimension of corporate social responsibility; CS, customer satisfaction; CI, customer identification; CL, customer loyalty; S.D, standard deviation; CR, composite reliability; AVE, average variance extracted; MaxR(H), maximum reliability.

The path model is shown in Figure 2 below. Table 3 below shows the direct effects between variables. The direct effect of sustainable development dimensions of CSR on customer satisfaction and customer loyalty was insignificant ($p > 0.05$), which indicates that the mediated model is superior to the direct model. Moreover, economic dimension of CSR is found to have a positive significant influence on customer identification (b = 0.254, $p < 0.05$). Thus, H1c was supported. The social and environmental dimension of CSR is found to be insignificant. Therefore, H1a and H1b were not supported. Customer identification has a significant positive influence on customer loyalty (b = 0.251, $p < 0.05$) and customer satisfaction (b = 0.218, $p < 0.05$).

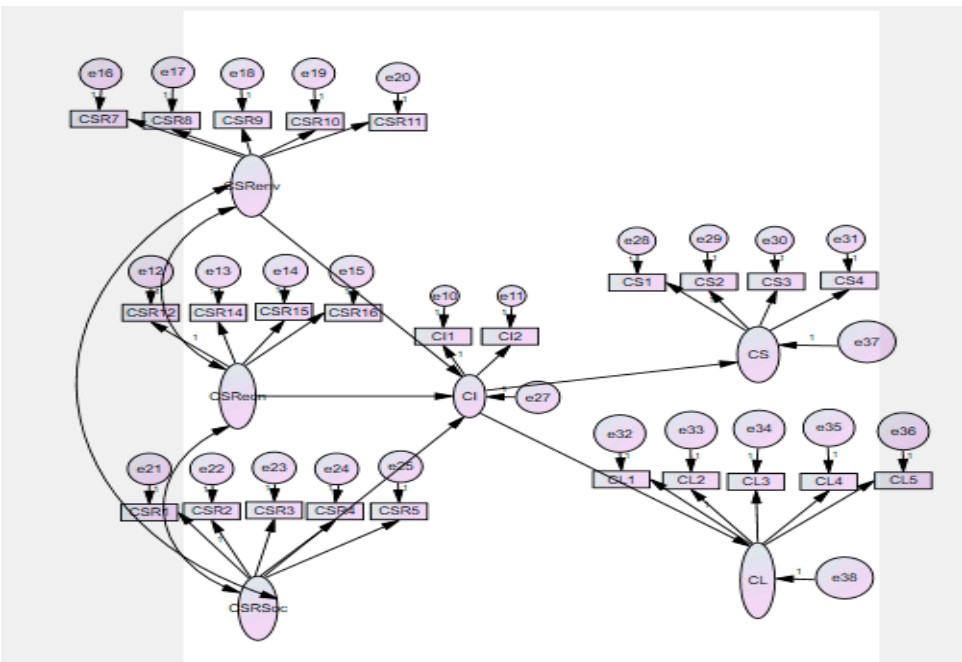

**Figure 2.** Path Model. Notes: CSRsoc, social dimension of corporate social responsibility; CSRecn, economic dimension of corporate social responsibility; CSRenv, environmental dimension of corporate social responsibility; CS, customer satisfaction; CI, customer identification; CL, customer loyalty.

**Table 3.** Direct path coefficients.

|  | Estimate | Lower | Upper | p | Decision |
|---|---|---|---|---|---|
| CSRenv→CI | −0.148 | −0.350 | 0.013 | 0.151 | H1b rejected |
| CSRecn→CI | 0.254 | 0.104 | 0.432 | *** | H1c supported |
| CSRsoc→CI | 0.118 | −0.088 | 0.387 | 0.379 | H1a rejected |
| CSRsoc→CL | 0.522 | −0.719 | 5.667 | 0.155 |  |
| CSRenv→CL | −0.380 | −4.804 | −0.120 | 0.119 |  |
| CSRecn→CL | 0.324 | −1.566 | 3.070 | 0.234 |  |
| CSRsoc→CS | 0.506 | −0.742 | 6.255 | 0.149 |  |
| CSRenv→CS | −0.366 | −5.148 | −0.115 | 0.219 |  |
| CSRecn→CS | 0.292 | −1.629 | 3.220 | 0.271 |  |
| CI→CL | 0.251 | 0.143 | 0.980 | *** |  |
| CI→CS | 0.218 | 0.105 | 0.896 | *** |  |

Notes: CSR, corporate social responsibility; CS, customer satisfaction; CI, customer identification; CL, customer loyalty; $p$, significance, ***, significant.

In order to test the mediation and indirect effect of customer identification, 95% bias-corrected bootstrapped confidence intervals ($n$ = 5000) were employed. The results are shown in Table 4 below. Customer identification is found to be a positive significant mediator between economic dimension of CSR and customer satisfaction (b = 0.071, $p < 0.05$). Thus, hypothesis 2c was supported. In addition, customer identification is found to be a significant positive mediator between the economic dimension of CSR and customer loyalty

(b = 0.075, $p < 0.05$), which supported hypothesis 3c. The mediating effect of customer identification was found to be insignificant for social and environmental dimensions of CSR and satisfaction and loyalty.

**Table 4.** Testing mediation effects.

| Parameter | Estimate | Lower | Upper | *p* | Decision |
|---|---|---|---|---|---|
| CSRenvxCIxCL | −0.037 | −0.144 | 0.004 | 0.162 | H3b rejected |
| CSRenvxCIxCS | −0.035 | −0.143 | 0.005 | 0.165 | H2b rejected |
| CSRecnxCIxCL | 0.075 | 0.028 | 0.339 | *** | H3c supported |
| CSRecnxCIxCS | 0.071 | 0.020 | 0.341 | *** | H2c supported |
| CSRsocxCIxCL | 0.028 | −0.020 | 0.342 | 0.389 | H3a rejected |
| CSRsocxCIxCS | 0.027 | −0.018 | 0.340 | 0.390 | H2a rejected |

Notes: CSR, corporate social responsibility; CS, customer satisfaction; CI, customer identification; CL, customer loyalty; *p*, significance, ***, significant.

## 5. Discussion

This study investigated the relationship between sustainable development dimensions of CSR and customer satisfaction and loyalty. Specifically, the mediation effect of customer identification has been tested. The results showed that the direct effect of CSR on the satisfaction and loyalty of customers is not significant, which indicated a mediating effect. This finding is similar to the study conducted by Raza, Rather [12]. However, the indirect effect of CSR is found to be significant. This finding can be explained by the cross-cultural differences between studies conducted in different countries and North Cyprus. In line with this finding, it is claimed that the benefits of CSR might not always be directly observed as customer loyalty and satisfaction as it can be observed through service quality [8,55]. It is supported that perceived CSR contributes to the emotions of the customers, such as identification [8]. The hierarchy of effect model proposed by [56] suggested that the customers follow stages when showing satisfaction or loyalty. This theory supports that different variables mediated this relationship, such as customer identification and emotions [8]. In line with the hierarchy of effect model, it is proven that customer identification fully mediates the relationship between CSR and customer satisfaction and loyalty. This implies the importance of establishing customer identification of customers with the company in order to observe customer-related outcomes.

Customers perceive the CSR of companies as implications of transparency, and honesty becomes more important in times of uncertainty. COVID-19 created uncertainty within the industries which enhanced the importance of CSR. Customers respond to the CSR-related image of companies by making decisions [57]. Specifically, the perceived CSR of a company affects the customer decisions such as identification, satisfaction, and loyalty to the company. Loyalty and feedback of customers tend to be positively affected by perceived CSR [44]. Customers are the key stakeholders that help the company to achieve its goals and objectives. CSR is proven to be a significant contributor to customer-related outcomes. Previous literature has provided some evidence that customer identifications with the organization influence customer loyalty [32]. The findings of the study supported the social identity theory and social categorization theory which suggested that customers can develop identification as a result of favorably perceived CSR activities of companies [38]. Customers tend to perceive the organizations' identity as their personal identity and, thus, tend to show favorable behaviors [32,48].

The findings of this study supported the importance of customer identification. CSR acts as a channel that signals the values of the company to the customers. Customers tend to identify themselves with the positively perceived companies. The identification is strengthened by the CSR [57]. Moreover, CSR is a key determinant of the perceived attractiveness of a company's identity that determines customer identification. Customers who are more strongly identified tend to have higher customer satisfaction. Previous studies have found that CSR is a direct contributor to customer satisfaction and customer loyalty. However, in contrast, this paper's findings suggest that CSR does not have a

direct effect on loyalty and satisfaction. Similarly, research conducted in the hospitality industry in the UK found the direct effect of CSR on loyalty and satisfaction to be insignificant [45]. However, social identity enhances the indirect relationship, which emphasizes the importance of achieving customer identification [8,13]. The previous findings and the findings of this study provided evidence that mediatory factors play an important role in this relationship. The economic dimension is found to be the only significant indirect contributor to customer loyalty and satisfaction. Currás-Pérez, Dolz-Dolz [23] suggested that the economic dimension of CSR is the most rational dimension of the sustainable development dimensions. Similar to this research, the economic dimension was found to influence customer functional or utilitarian value [23]. In the long term, customers perceive firms to be more successful in economic performance that creates value for customers through products/services [23]. Social and environmental dimensions did not show an effect on the customers. This can be explained by the cross-cultural differences between respondents of other studies. In their research [58] suggested that there is a difference between perceptions of customers from different countries. It was concluded that Spanish customers were not as concerned about the environmental issues as the UK customers [58].

In addition, banks are one the institutions with the highest contribution to CSR activities, making them key for the CSR literature [33]. CSR activities in developing countries show contrast with those in developed countries [18]. The findings of this research provided evidence that CSR does not have a direct effect on customer satisfaction and loyalty in the absence of customer identification. This has emphasized the importance of identification for developing countries. Moreover, the findings suggested that managers should focus on developing customer–company identification. In that sense, identification can be managed by managers, and CSR is a tool to enhance identification which can contribute to long-term loyalty and satisfaction. Although engaging in CSR initiatives is a costly concept for banks, the findings of this study indicated that it is crucial. It should be implemented within the strategies of banks, considering its benefits in achieving customer-related outcomes through developing customer identification. In the long term, the cost of CSR initiatives is likely to result in increased profitability for banks.

## 6. Conclusions

This study aimed to analyze the relationship between a sustainable development perspective of CSR and customer loyalty and satisfaction through the mediation of customer identification. It provides invaluable insights into the developing country and banking industry perspective on the proposed relationship. The direct effect of sustainable development dimensions of CSR was found to be insignificant on the customer satisfaction; however, the indirect effects of the economic dimension of CSR were significant, which suggests a full mediation effect. The importance of engaging in CSR initiatives and achieving customer identification were emphasized. The findings of this study are expected to provide a road map for bank managers in developing countries in order to for long-term success. In order to observe customer-related benefits, banks must supply insightful information to the stakeholders, including customers. Furthermore, both in the current study and in the literature, it is widely acknowledged that CSR activities are perceived to be important for the customers. This can help to build a connection built on trust and loyalty in this manner [28]. In order to achieve a sustainable customer–company relationship in the case of developing countries, banks should adopt CSR strategies. Even though banks are expected to engage in CSR initiatives within a normative perspective, there are certain benefits of conformity with the CSR expectations of customers, such as loyalty and satisfaction [59,60]. In developing countries where circumstances are challenging, banks' role in contributing to the well-being of society becomes more crucial. CSR initiatives are a way of gaining the trust of bank customers. Furthermore, it is suggested that in order to benefit the customers and the community, both high-level and branch-level managers should be involved in the bank's CSR strategy and empowered to engage in CSR activities that benefit their customers and the community [43].

There are certain limitations of this study. The study was focused on the banking industry, which limits the generalizability of the findings to other industries. Thus, future studies can focus on different industries, which can enable the generalization and comparison of the findings. In addition, this study is conducted in North Cyprus, which is a developing country. A study can be conducted in developed countries and in a different geographical context which can also enhance the generalization. Furthermore, customer identification was used as the mediating variable considering relevant theories. Different mediating variables such as customer trust and theoretical basis can be tested to further contribute to the literature. Lastly, quantitative research techniques can be supported by qualitative research techniques, including in-depth interviews, which can provide further support for the findings.

**Author Contributions:** Supervision S.Z.E.; Writing—original draft M.O.; Writing—review & editing K.C. All authors have read and agreed to the published version of the manuscript.

**Funding:** This research received no external funding.

**Institutional Review Board Statement:** The study was conducted in accordance with the Declaration of Helsinki, and approved by the Ethics Committee of NEU Scientific Research Ethics Committee (YDÜ/SB/2020/871).

**Informed Consent Statement:** Informed consent was obtained from all subjects involved in the study.

**Data Availability Statement:** Data sharing is not applicable.

**Conflicts of Interest:** The authors declare no conflict of interest.

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
