# Peer review of "Sustainable Development and Customer Satisfaction and Loyalty in North Cyprus: The Mediating Effect of Customer Identification"

_sustainability, doi:10.3390/su14095196_

Round 1

Reviewer 1 Report

some feedback for the authors:

Title and Introduction

  1. The author should link the statements that most studies about CSR are in developed countries and the position of northern Cyprus. Readers should be notified if Northern Cyprus is part of developing countries or developed nations.
  2. Since this study investigated the CSR initiative in the developing country (Cyprus), the title should be stress on the evidence from the country

Theoretical framework and Hypotheses

The structure of theoretical framework and hypotheses development should be restructured based on the subject or hypotheses. For example, section 2.2 should be CSR and customer identification and so on.

Since the focus of the respondents used in this article is banking sector customers, there should be a review that discusses the implication of CSR in banking sectors in literature review section.

The application of CSR in the other practices could be considered discussed in the first section of 2.1 (CSR). For example, the relevant study about the impact of CSR in global supply chain business can be considered in this study http://www.ijbs.unimas.my/index.php/content-abstract/all-issues/63-vol-19-no-3-2018/509-corporate-social-responsibility-in-supply-chains-the-impact-in-the-context-of-global-supply-chains.  

Methodology

The final paragraph of the mythology section that shows the profile of respondents should be in the results section.

In methodology does not mention how the respondents are targeted and there is no information what are the criteria of bank industry need to be investigated in this study. Small bank and well established bank have different amount of CSR.

what kind of statistical approaches used in this study

  1. Results.

There is no model or framework results from statistical tool used in the result section that shows the relationship between variables. Please add them in the results section

  1. Discussion

The discussion should be stressed in the connection of the findings and respondent demographic data.

Discussing the finding and its relation to the developing or developed countries seems not relevant. Please consider other perspective.

Author Response

Reviewer’s Comments

The author should link the statements that most studies about CSR are in developed countries and the position of northern Cyprus. Readers should be notified if Northern Cyprus is part of developing countries or developed nations.

Author’s Response

The statements are linked and North Cyprus is emphasized as a developing country.

Reviewer’s Comments

Since this study investigated the CSR initiative in the developing country (Cyprus), the title should be stress on the evidence from the country

Author’s Response

Title have been updated to include North Cyprus.

Reviewer’s Comments

The structure of theoretical framework and hypotheses development should be restructured based on the subject or hypotheses. For example, section 2.2 should be CSR and customer identification and so on.

Author’s Response

The sections are structured accordingly.

Reviewer’s Comments

Since the focus of the respondents used in this article is banking sector customers, there should be a review that discusses the implication of CSR in banking sectors in literature review section.

Author’s Response

Banking sector and csr is discussed within the hypothesis development sections when developing the hypothesis.

Reviewer’s Comments

The application of CSR in the other practices could be considered discussed in the first section of 2.1 (CSR). For example, the relevant study about the impact of CSR in global supply chain business can be considered in this study http://www.ijbs.unimas.my/index.php/content-abstract/all-issues/63-vol-19-no-3-2018/509-corporate-social-responsibility-in-supply-chains-the-impact-in-the-context-of-global-supply-chains.  

Author’s Response

The relevant study is considered and other practices have been discussed.

Reviewer’s Comments

The final paragraph of the mythology section that shows the profile of respondents should be in the results section.

Author’s Response

It is moved to the results section.

Reviewer’s Comments

In methodology does not mention how the respondents are targeted and there is no information what are the criteria of bank industry need to be investigated in this study. Small bank and well established bank have different amount of CSR.

Author’s Response

Local banks in North Cyprus have similar sizes in North Cyprus. Some international banks have been excluded as they have different backgrounds and sizes. Local banks that have at least a branch in each city of North Cyprus is selected.

Reviewer’s Comments

what kind of statistical approaches used in this study

Author’s Response

Quantitative approaches have been used.

Reviewer’s Comments

There is no model or framework results from statistical tool used in the result section that shows the relationship between variables. Please add them in the results section

Author’s Response

Path model have been added to the results section.

Reviewer’s Comments

The discussion should be stressed in the connection of the findings and respondent demographic data. Discussing the finding and its relation to the developing or developed countries seems not relevant. Please consider other perspective.

Author’s Response

Sustainable development perspective of CSR and previous findings of the related literature have been discussed and emphasized in the discussion section as well as the developing countries case.

Reviewer 2 Report

Dear authors,

While I found the topic addressed interesting, my overall assessment is that the manuscript has several issues, in particular the contribution to the field (unclear and not sufficiently justified) and the discussion of the results (lack of depth).

Please find my detailed comments below.

  1. The contribution to the field must be clearly explained. On page 2, lines 63-66, the authors state that “It is criticized that studies have focused on loyalty and satisfaction in the previous literature in separate models, as it is suggested that these two parameters should be considered in the same conceptual model (Martinez et al., 2014).” However, previous studies have already focused on customer satisfaction and loyalty in the same conceptual model (e.g., Al-Ghamdi & Badawi, 2019; Moliner et al., 2020). Moreover, some previous studies also examined the influence of corporate social responsibility on customer satisfaction and/or customer loyalty through the mediating effect of customer identification (e.g., Pérez & Rodríguez del Bosque, 2015, 2016; Raza, Saeed, et al., 2020). Thus, what is the significant difference between this paper and previous literature? Which previous gap tries to solve? Please consider the following references not only to clarify the contribution to the field but also to update the literature review.

Abbas, M., Gao, Y., & Shah, S. S. H. (2018). CSR and customer outcomes: The mediating role of customer engagement. Sustainability, 10(11). doi:10.3390/su10114243

Al-Ghamdi, S. A., & Badawi, N. S. (2019). Do corporate social responsibility activities enhance customer satisfaction and customer loyalty? Evidence from the Saudi banking sector. Cogent Business & Management, 6(1). doi:10.1080/23311975.2019.1662932

Ltifi, M., & Hichri, A. (2022). The effects of corporate governance on the customer’s recommendations: A study of the banking sector at the time of COVID-19. Journal of Knowledge Management, 26(1), 165-191. doi:10.1108/JKM-06-2020-0471

Moliner, M. A., Monferrer Tirado, D., & Estrada-Guillén, M. (2020). CSR marketing outcomes and branch managers’ perceptions of CSR. International Journal of Bank Marketing, 38(1), 63-85. doi:10.1108/IJBM-11-2018-0307

Pérez, A., & Rodríguez del Bosque, I. (2015). An integrative framework to understand how CSR affects customer loyalty through identification, emotions and satisfaction. Journal of Business Ethics, 129(3), 571-584. doi:10.1007/s10551-014-2177-9

Pérez, A., & Rodríguez del Bosque, I. (2016). The stakeholder management theory of CSR: A multidimensional approach in understanding customer identification and satisfaction. International Journal of Bank Marketing, 34(5), 731-751. doi:10.1108/IJBM-04-2015-0052

Raza, A., Rather, R. A., Iqbal, M. K., & Bhutta, U. S. (2020). An assessment of corporate social responsibility on customer company identification and loyalty in banking industry: A PLS-SEM analysis. Management Research Review, 43(11), 1337-1370. doi:10.1108/MRR-08-2019-0341

Raza, A., Saeed, A., Iqbal, M. K., Saeed, U., Sadiq, I., & Faraz, N. A. (2020). Linking corporate social responsibility to customer loyalty through co-creation and customer company identification: Exploring sequential mediation mechanism. Sustainability, 12(6). doi:10.3390/su12062525

Uslu, A., & Åžengün, H. Ä°. (2021). The multiple mediation roles of trust and satisfaction in the effect of perceived corporate social responsibility on loyalty. Business, Management and Economics Engineering, 19(1), 49-69. doi:10.3846/bmee.2021.13362

  1. On page 2, lines 69-71, the authors state that “In the context of Northern Cyprus, where the current study was carried out, there is lack of research on the CSR and customer specific outcomes that focus on the banking industry (….)”. What are the particularities of banks that make them unique and justify a separate analysis? Additional arguments related to the nature of the banking activity must be added to support the research.

  1. On subsection 2.2. the authors develop the research hypotheses. In order to improve the visualization of the conceptual model, I suggest that the authors provide, as is usual in the literature, an explanatory figure to illustrate the proposed model.

  1. On page 6:

- Line 290, the authors refer that “(…) the discriminant validity was checked by considering the AVE (…)”. However, AVE measures convergent validity as, also, the authors state on page 7 lines 301-302 “AVE should be higher than 0.50 in order to achieve convergent validity!”. Please revise.

- Table 1: the definitions of the scale items must be provided. Please also revise the first column (CL and CS and not CA).

  1. On page 7, lines 311-312, the authors state that “The direct effect of CSR on customer satisfaction and customer loyalty were insignificant (p>0.05) (…)” and on page 8, lines 335-337, they refer that “The results showed that the direct effect of CSR on the satisfaction and loyalty of customers is not significant.” However, the authors do not provide (e.g., Table 3), the coefficients of the direct effect of CSR on customer satisfaction and customer loyalty. Please present such coefficients.

  1. The discussion of results is too descriptive rather than analytical. Thus, they leave the question "so what?" not adequately answered. It should be noted that the discussion of results is critical as it allows not only to relate the study findings to previous studies but mainly to explain any new understanding or insights that emerged from the research developed (i.e., strengthen the contribution of the research).

  1. On page 9, lines 363-364, the authors stress that “However, In contrast this paper’s findings suggest that CSR does not have a direct effect on the loyalty and ” The authors should provide possible explanations for the differences between their results and those reported in previous studies.

  1. Both theoretical and practical implications must be presented and explained in detail. Mention that “The findings of this study are expected to provide a road map for bank managers in developing countries in order to for long-term success” (page 9, lines 392-393) is clearly insufficient.

  1. List (not exhaustive) of typos and other issues:

- Page 2, line 46: “(…) has shown that CSR can lead (…)” and not “(…) has shown that that corporate social responsibility can lead (…)”.

- Page 2, lines 47-48: “(…) companies with CSR (…)” and not “(…) companies with corporate social responsibility (…)”.

- Page 2, line 95: “(…) in all explanations (…)” and not “(…) inall explanations (…)”.

- Page 5, line 234: “(…) to test the hypotheses in (…)” and not “(…) to test thehypothesis in (…)”.

- Page 9, line 361: “Customers who are more (…)” and not “Customers whoare more (…)”.

Author Response

Reviewer’s Comments

The contribution to the field must be clearly explained. On page 2, lines 63-66, the authors state that “It is criticized that studies have focused on loyalty and satisfaction in the previous literature in separate models, as it is suggested that these two parameters should be considered in the same conceptual model (Martinez et al., 2014).” However, previous studies have already focused on customer satisfaction and loyalty in the same conceptual model (e.g., Al-Ghamdi & Badawi, 2019; Moliner et al., 2020). Moreover, some previous studies also examined the influence of corporate social responsibility on customer satisfaction and/or customer loyalty through the mediating effect of customer identification (e.g., Pérez & Rodríguez del Bosque, 2015, 2016; Raza, Saeed, et al., 2020). Thus, what is the significant difference between this paper and previous literature? Which previous gap tries to solve? Please consider the following references not only to clarify the contribution to the field but also to update the literature review.

Abbas, M., Gao, Y., & Shah, S. S. H. (2018). CSR and customer outcomes: The mediating role of customer engagement. Sustainability, 10(11). doi:10.3390/su10114243

Al-Ghamdi, S. A., & Badawi, N. S. (2019). Do corporate social responsibility activities enhance customer satisfaction and customer loyalty? Evidence from the Saudi banking sector. Cogent Business & Management, 6(1). doi:10.1080/23311975.2019.1662932

Ltifi, M., & Hichri, A. (2022). The effects of corporate governance on the customer’s recommendations: A study of the banking sector at the time of COVID-19. Journal of Knowledge Management, 26(1), 165-191. doi:10.1108/JKM-06-2020-0471

Moliner, M. A., Monferrer Tirado, D., & Estrada-Guillén, M. (2020). CSR marketing outcomes and branch managers’ perceptions of CSR. International Journal of Bank Marketing, 38(1), 63-85. doi:10.1108/IJBM-11-2018-0307

Pérez, A., & Rodríguez del Bosque, I. (2015). An integrative framework to understand how CSR affects customer loyalty through identification, emotions and satisfaction. Journal of Business Ethics, 129(3), 571-584. doi:10.1007/s10551-014-2177-9

Pérez, A., & Rodríguez del Bosque, I. (2016). The stakeholder management theory of CSR: A multidimensional approach in understanding customer identification and satisfaction. International Journal of Bank Marketing, 34(5), 731-751. doi:10.1108/IJBM-04-2015-0052

Raza, A., Rather, R. A., Iqbal, M. K., & Bhutta, U. S. (2020). An assessment of corporate social responsibility on customer company identification and loyalty in banking industry: A PLS-SEM analysis. Management Research Review, 43(11), 1337-1370. doi:10.1108/MRR-08-2019-0341

Raza, A., Saeed, A., Iqbal, M. K., Saeed, U., Sadiq, I., & Faraz, N. A. (2020). Linking corporate social responsibility to customer loyalty through co-creation and customer company identification: Exploring sequential mediation mechanism. Sustainability, 12(6). doi:10.3390/su12062525

Uslu, A., & Åžengün, H. Ä°. (2021). The multiple mediation roles of trust and satisfaction in the effect of perceived corporate social responsibility on loyalty. Business, Management and Economics Engineering, 19(1), 49-69. doi:10.3846/bmee.2021.13362

Author’s Response

The contribution of the paper is made clear considering the suggested literatures. These references have also been used in the literature review and discussion parts of the paper.

 Reviewer’s Comments

On page 2, lines 69-71, the authors state that “In the context of Northern Cyprus, where the current study was carried out, there is lack of research on the CSR and customer specific outcomes that focus on the banking industry (….)”. What are the particularities of banks that make them unique and justify a separate analysis? Additional arguments related to the nature of the banking activity must be added to support the research.

 Author’s Response

The importance of banks, banking sector and North Cyprus have been included in the introduction, discussion and conclusion parts of the study.

 Reviewer’s Comments

On subsection 2.2. the authors develop the research hypotheses. In order to improve the visualization of the conceptual model, I suggest that the authors provide, as is usual in the literature, an explanatory figure to illustrate the proposed model.

Author’s Response

A figure is induced at the end of the literature review section. A path model have also been included in the results section.

  Reviewer’s Comments

- Line 290, the authors refer that “(…) the discriminant validity was checked by considering the AVE (…)”. However, AVE measures convergent validity as, also, the authors state on page 7 lines 301-302 “AVE should be higher than 0.50 in order to achieve convergent validity!”. Please revise.

- Table 1: the definitions of the scale items must be provided. Please also revise the first column (CL and CS and not CA).

Author’s Response

The validity issues have been corrected. Table 1 is also corrected.

 Reviewer’s Comments

On page 7, lines 311-312, the authors state that “The direct effect of CSR on customer satisfaction and customer loyalty were insignificant (p>0.05) (…)” and on page 8, lines 335-337, they refer that “The results showed that the direct effect of CSR on the satisfaction and loyalty of customers is not significant.” However, the authors do not provide (e.g., Table 3), the coefficients of the direct effect of CSR on customer satisfaction and customer loyalty. Please present such coefficients.

Author’s Response

Direct effects have been included in the table 3.

Reviewer’s Comments

The discussion of results is too descriptive rather than analytical. Thus, they leave the question "so what?" not adequately answered. It should be noted that the discussion of results is critical as it allows not only to relate the study findings to previous studies but mainly to explain any new understanding or insights that emerged from the research developed (i.e., strengthen the contribution of the research).

Author’s Response

The contribution of the research is improved in the introduction section and also the discussion section is improved accordingly.

Reviewer’s Comments

On page 9, lines 363-364, the authors stress that “However, In contrast this paper’s findings suggest that CSR does not have a direct effect on the loyalty and ” The authors should provide possible explanations for the differences between their results and those reported in previous studies.

Author’s Response

Differences with previous studies have been added considering the previous findings about the direct and indirect effects.

Reviewer’s Comments

Both theoretical and practical implications must be presented and explained in detail. Mention that “The findings of this study are expected to provide a road map for bank managers in developing countries in order to for long-term success” (page 9, lines 392-393) is clearly insufficient.

Author’s Response

The concluding part is improved to give more insights for banks and bank managers about CSR.

 Reviewer’s Comments

List (not exhaustive) of typos and other issues:

- Page 2, line 46: “(…) has shown that CSR can lead (…)” and not “(…) has shown that that corporate social responsibility can lead (…)”.

- Page 2, lines 47-48: “(…) companies with CSR (…)” and not “(…) companies with corporate social responsibility (…)”.

- Page 2, line 95: “(…) in all explanations (…)” and not “(…) inall explanations (…)”.

- Page 5, line 234: “(…) to test the hypotheses in (…)” and not “(…) to test thehypothesis in (…)”.

- Page 9, line 361: “Customers who are more (…)” and not “Customers whoare more (…)”.

Author’s Response

The typos have been corrected.

Reviewer 3 Report

Dear authors,
The content of the study closely corresponds to the title, and the problems discussed are presented in a clear and transparent manner, which makes them interesting for the reader. The paper is well structured, however, there are some minor adjustments to be taken into account.

  • An abstract should be structured as follows: the aim of the paper, methods, results, conclusions, and recommendations/future directions. The elements of novelty are not clear, it is worth supplementing.
  • The Authors focused on three dimensions of CSR - economic, social and environmental. They should explain why these 3 areas of CSR were chosen. It is also worth supplementing it in the literature review.
  • It is worth supplementing the theoretical framework with more current (mostly within the last 5 years) references to this subject.

Author Response

Reviewer’s Comments

An abstract should be structured as follows: the aim of the paper, methods, results, conclusions, and recommendations/future directions. The elements of novelty are not clear, it is worth supplementing.

Author’s Response

Abstract is updated in accordance with the reviewer’s comments. The novelty is also emphasized considering the sustainable development perspective.

Reviewer’s Comments

The Authors focused on three dimensions of CSR - economic, social and environmental. They should explain why these 3 areas of CSR were chosen. It is also worth supplementing it in the literature review.

Author’s Response

The three dimensions have been explained in the literature review section and their importance is emphasized by supplementing with related references.

Reviewer’s Comments

It is worth supplementing the theoretical framework with more current (mostly within the last 5 years) references to this subject.

Author’s Response

The study is supported with recent references.

Reviewer 4 Report

Thank you very much for giving me the opportunity to read this interesting study. Please consider following observations:

The introduction part is not successful in introducing the problem background, motivation and significance of the research. Revise it.

In literature review, phenomenon should be explained through the help of some prevailing theories , which could provide scientific basis of the hypotheses. For better understanding, read following papers. 

“Does whipping Tournament Incentives spur CSR Performance? An empirical evidence from Chinese Subnational Institutional Contingencies” Frontiers in Psychology, (2022)  10.3389/fpsyg.2022.841163

“Board Composition and Social & Environmental Accountability: A Dynamic Model Analysis of Chinese Firms” Sustainability (2021) 13(19), 10662

Literature review should be augmented by freshly published studies. 

Conclusion should possess limitations and practical and theoretical implications. 

Author Response

Reviewer’s Comments

The introduction part is not successful in introducing the problem background, motivation and significance of the research. Revise it.

Author’s Response

Introduction part is improved to indicated the contributions of the study.

Reviewer’s Comments

In literature review, phenomenon should be explained through the help of some prevailing theories , which could provide scientific basis of the hypotheses. For better understanding, read following papers.  “Does whipping Tournament Incentives spur CSR Performance? An empirical evidence from Chinese Subnational Institutional Contingencies” Frontiers in Psychology, (2022)  10.3389/fpsyg.2022.841163’’ ‘’Board Composition and Social & Environmental Accountability: A Dynamic Model Analysis of Chinese Firms” Sustainability (2021) 13(19), 10662’’

Author’s Response

Prevailing theories have been added to the literature review and to the discussion section.

Reviewer’s Comments

Literature review should be augmented by freshly published studies. 

Author’s Response

Recent studies have been added to the references of the study.

Reviewer’s Comments

Conclusion should possess limitations and practical and theoretical implications.

Author’s Response

Limitations and implications have been added further to the conclusion section.

Round 2

Reviewer 2 Report

Dear Authors,

The revised manuscript meets the conditions for publication.

Reviewer 4 Report

Thank you for incorporating my comments.